# Improving the precision of modeling the incidence of hemorrhagic fever with renal syndrome in mainland China with an ensemble machine learning approach

**Guo-hua Ye[1], Mirxat Alim[1], Peng Guan[1], De-sheng Huang[2], Bao-sen Zhou[1], Wei Wu[1]***

**1** Department of Epidemiology, School of Public Health, China Medical University, Shenyang, Liaoning, China, **2** Department of Mathematics, School of Fundamental Sciences, China Medical University, Shenyang, Liaoning, China

* wuwei@cmu.edu.cn

**Data Availability Statement:** All relevant data are within the paper and its Supporting Information files.

## Abstract

### Objective

Hemorrhagic fever with renal syndrome (HFRS), one of the main public health concerns in mainland China, is a group of clinically similar diseases caused by hantaviruses. Statistical approaches have always been leveraged to forecast the future incidence rates of certain infectious diseases to effectively control their prevalence and outbreak potential. Compared to the use of one base model, model stacking can often produce better forecasting results. In this study, we fitted the monthly reported cases of HFRS in mainland China with a model stacking approach and compared its forecasting performance with those of five base models.

### Method

We fitted the monthly reported cases of HFRS ranging from January 2004 to June 2019 in mainland China with an autoregressive integrated moving average (ARIMA) model; the Holt-Winter (HW) method, seasonal decomposition of the time series by LOESS (STL); a neural network autoregressive (NNAR) model; and an exponential smoothing state space model with a Box-Cox transformation; ARMA errors; and trend and seasonal components (TBATS), and we combined the forecasting results with the inverse rank approach. The forecasting performance was estimated based on several accuracy criteria for model prediction, including the mean absolute percentage error (MAPE), root-mean-squared error (RMSE) and mean absolute error (MAE).

### Result

There was a slight downward trend and obvious seasonal periodicity inherent in the time series data for HFRS in mainland China. The model stacking method was selected as the best approach with the best performance in terms of both fitting (RMSE 128.19, MAE 85.63, MAPE 8.18) and prediction (RMSE 151.86, MAE 118.28, MAPE 13.16).

**Funding:** This study was supported by the National Natural Science Foundation of China (Grant No. 81202254 and 71974199) and the Health and Medical Big Data Research Project of China Medical University (Grant No. HMB201903105).

**Competing interests:** The authors have declared that no competing interests exist.

## Conclusion

The results showed that model stacking by using the optimal mean forecasting weight of the five abovementioned models achieved the best performance in terms of predicting HFRS one year into the future. This study has corroborated the conclusion that model stacking is an easy way to enhance prediction accuracy when modeling HFRS.

## 1. Introduction

Hemorrhagic fever with renal syndrome (HFRS) is an infectious rodent-borne disease caused by hantavirus, and it is a significant public health problem in China, comprising approximately 90% of HFRS cases reported globally [1]. In China, HFRS is predominantly caused by two strains of hantavirus, the Hantaan virus and Seoul virus, and its clinical symptoms include fever, bleeding and acute kidney injury [2]. Humans become infected with hantavirus mainly through exposure to aerosols with contaminated urine and feces, ingestion of contaminated food and rodent bites [3–5]. It has been conclusively proven that the incidence of HFRS contagion is contingent upon the population densities of rodents, the level of exposure, and other ambient factors, such as temperature, humidity and the level of urbanization [6]. In a countermeasure designed to stop the spread of HFRS in China, the Chinese Center for Disease Control and Prevention (CDC) contrived a comprehensive mandatory surveillance system to collect data related to HFRS and publicize this data on its official website. A statistical approach for handling data pertaining to HFRS has been routine practice to describe its epidemiological characteristics and predict its occurrence characteristics in the future by modeling the data.

There have been many reports that are solely focused on time series analyses of the incidence rate of HFRS; for example, Liu et al. predicted the incidence rate of HFRS three years into the future through the autoregressive integrated moving average (ARIMA) method based on yearly incidence rate data ranging from 1978 to 2008, and the model did not show significant autocorrelation in the residual [7]. Wang et al. predicted the incidence rate of HFRS twelve months into the future using the ARIMA method based on monthly incidence rate data ranging from 2004 to 2013, and this approach produced reliable results [8]. Other researchers have resorted to more complicated methods. Wu et al. used both the ARIMA method and a NAR neural network to model monthly incidence data for HFRS ranging from 2004 to 2013, and they determined that the NAR neural network method outperforms the ARIMA method in fitting and predicting the incidence tendency of HFRS [9].

In our previous study [10], we compared the performances of two hybrid models in fitting HFRS incidence data from Jiangsu Province, China. One hybrid model was composed of a nonlinear autoregressive neural network (NARNN) and ARIMA, and the other was composed of a generalized regression neural network (GRNN) and ARIMA. In contrasted to this study in terms of the employed model mechanism, in the hybrid model study, we first fitted the raw data with ARIMA, after which we tried to obtain additional potentially available information from the residual with the help of a neural network. In this study, we fit the data with five different models and then combine the forecasting result by using the inverse rank approach to obtain the best possible performance. Therefore, it can be seen that this study and our previous study adopt two different strategies.

The aforementioned studies focused on either a single model or several models and comparisons between them, and few studies have examined the combination of the outcomes of different models. Previous studies have shown that model stacking often leads to better

accuracy than that of a single model by integrating several base models to yield a final result. Model stacking has been used in many fields to enhance the precision of prediction [11–13]. In many cases, this would lead to significant improvements in accuracy either by using the averaging approach or weighted averaging approach. In this study, we not only construct and compare five base models to fit the monthly reported incidence of HFRS, but also combine the forecast outcomes of those models to achieve improved prediction results.

## 2. Materials and method

### 2.1 Data collection and preprocessing

The monthly reported cases of HFRS ranging from January 2004 to June 2019 was obtained from the official website of National Health Commission of the People's Republic of China (http://www.nhc.gov.cn/). The dataset analyzed during the study is included in S1 Dataset. In China, HFRS is a statutorily notifiable disease, and hospitals must report every case to their local health authority within 24 hours. Each HFRS case was diagnosed and confirmed by clinical symptoms and serological identification, respectively. The incidence data collected from the HFRS surveillance system contain no personal information; therefore, this study does not require an ethics examination. The data from January 2004 to June 2018 are used as a training dataset and the remaining data (from July 2018 to June 2019) are employed as a testing dataset to check the forecasting performance of the developed model. All statistical analyses are performed with the R 3.6.1 software. To preliminarily acknowledge the basic characteristics of the data, we performed seasonal decomposition to determine the existence of seasonal components. A Box-Cox transformation is required in cases where normalization is necessary.

### 2.2 Base models and model stacking

**2.2.1 ARIMA.** The ARIMA method has often been used for the prediction of infectious diseases [14–16], such as the HFRS. Here, we construct a seasonal ARIMA (p, d, q) (P, D, Q) [S] model, because of the seasonality of the monthly incidence data, in which the p parameter is the order of autoregression, d is the order of differencing, q is the order of the moving average, and S is the period of seasonality. (P, D, Q) denotes the seasonal component of the model. The original data must be converted to stationary time series by differencing or other approaches. The six parameters are chosen by the criteria of the partial autocorrelation function (PACF) and autocorrelation function (ACF). S is decided by the periodic length of the seasonality, and finally, the optimal model is selected based on the Akaike information criterion (AIC). Additionally, the Ljung-Box test is used to examine the homogeneity of the residuals produced by the ARIMA models to assess their fitness values [17].

**2.2.2 Holt-Winters seasonal method (Holt-Winters additive model with additive error and no trend).** The Holt-Winters(HW) method is a member of the exponential smoothing model family that was proposed by Holt [18] and Winters [19]. The HW seasonal method is tailored to directly capture the seasonality of time series data by assuming that the predicted data and historical data share some common iterative data-generation patterns, namely, the estimation of future values is based on past data. The model contains a forecasting equation and three smoothing equation, each of which is used for different components; one for level, one for trend and one for seasonality. In contrast with ARIMA, the optimal HW model is produced by iterative calculation instead of approximation to the statistical model. There are two types of HW methods that differ in terms of the characteristics of their seasonal components; one is multiplicative, and the other is additive. The multiplicative method is favored when the seasonal component changes proportionally with the level of the time series. The additive

method is favored when the seasonal component is relatively constant throughout the entire time series. The two different methods are described mathematically as follows:

Multiplicative HW model:

$$\text{Level:}\ L_t = \alpha\left(\frac{Y_t}{S_{t \cdot m}}\right) + [1 - \alpha] \times (L_{t-1} - b_{t-1}) \tag{1}$$

$$\text{Trend:}\ b_t = \beta \times (L_t L_{t-1}) + (1 - \beta) \times b_{t-1} \tag{2}$$

$$\text{Seasonality:}\ S_t = \gamma \times \left(\frac{Y_t}{L_t}\right) + (1 - \gamma) \times b_{t-1} \tag{3}$$

$$\text{Forecast equation:}\ F_{t+k} = (L_t + k \times b_t) \times S_{t+k-m} \tag{4}$$

Additive HW model:

$$\text{Level:}\ L_t = \alpha(Y_t - S_{t \cdot m}) + [1 - \alpha] \times (L_{t-1} - b_{t-1}) \tag{5}$$

$$\text{Trend:}\ b_t = \beta \times (L_t L_{t-1}) + (1 - \beta) \times b_{t-1} \tag{6}$$

$$\text{Seasonality:}\ S_t = \gamma \times (Y_t - L_t) + (1 - \gamma) \times b_{t-1} \tag{7}$$

$$\text{Forecast equation:}\ F_{t+k} = (L_t + k \times b_t) \times S_{t+k-m} \tag{8}$$

M is used to denote the periodicity of the seasonality, t refers to the time stamp of certain recording, k is the index, and $\alpha$, $\beta$ and $\gamma$ are smoothing factors that are theoretically limited between 0 and 1. The final optimal model is obtained by minimizing the squared one-step prediction error [20].

**2.2.3 STL.** STL is a versatile and robust approach for decomposing time series; it is an acronym that represents 'seasonal and trend decomposition using LOESS (locally-weighted scatterplot smoothing)', and LOESS is a method specifically designed for fitting nonlinear relationships [21]. The STL method assumes that the time series data include three components: a trend component, a seasonal component and the remainder of the data. The STL method has the advantage of allowing the seasonal component to change over time, and the rate of change and the smoothness of the trend period can be manipulated manually. The performance can be robust without considering outliers, so an occasional rare variation does not affect the holistic estimation. The algorithm first identifies the trend component and then calculates the seasonal component. Decomposition with either the additive or multiplicative method can be performed by using a Box-Cox transformation with the $\lambda$ parameter. When $\lambda = 0$, the multiplicative decomposition is performed, while $\lambda = 1$ denotes additive decomposition.

**2.2.4 NNAR.** Neural network techniques have long been widely used for the forecasting of infectious diseases [22]. NNAR is an acronym for 'neural network auto regressive'. The NNAR model is a three-layer feedforward neural network with only one hidden layer and lagged input for forecasting univariate time series. The relationship between the input and output can be represented by the following equation:

$$y_t = w_0 + \sum_{j=1}^{h} w_j \cdot g\left(w_{0,j} + \sum_{i=1}^{n} w_{i,j} \cdot y_{t-1}\right) + \varepsilon_t \tag{9}$$

where w are the parameters, n refers to the number of input nodes and h represents the number of hidden nodes. The hidden layer transfer function is a sigmoid function, and the

activation function for the output is a linear function. In this study, we choose the NNAR (p, P, k)[m] model. p refers to the number of nonseasonal inputs, P refers to the number of seasonal inputs, k refers to the number of hidden nodes, and m denotes the cycle length of seasonality.

An important preliminary task is to determine the best values of p and P, and this should be done before fitting the NNAR model. A normalization of the time series is required before plotting the PACF, which determines the number of nonseasonal lags. The P value is decided according to the seasonal pattern of the original monthly incidence data.

**2.2.5 TBATS.** TBATS is a conglomerate of the trigonometric seasonal model, a Box-Cox transformation and the trend and seasonal decompositions with errors fitted by the ARIMA method [23, 24]. The mathematical procedure of the TBATS method is extremely intricate and out of the purview of this study. Generally, the training data are primarily decomposed into trend, seasonal and residual components. Then, a Box-Cox transformation is used to address problems related to nonlinearity. The error component is fitted with the ARIMA model, while the trigonometric function is utilized to address the seasonal non-integer periodicity [25]. A damping parameter is necessary to restrict the continuity of the trend component. The final optimal model is obtained based on the minimization of the AIC via an iterative process. The final model is described as follow:

$$\text{TBAT}(\omega, \varphi, p, q, \{m_1, k_1\}, \ldots \ldots \ldots \{m_T k_T\}) \tag{10}$$

where p, q are the parameters of ARIMA, and $\omega$ and $\varphi$ are the Box-Cox transformation parameter and damping parameter, respectively. m and k refer to the seasonal periodicity and relevant parameter applied to each seasonality, respectively.

**2.2.6 Model stacking.** The end result is acquired by adding different weights to each of the means of the five base models described above. The weights are calculated according to the inverse rank approach proposed by Aiofli and Timmermann [26]. Briefly, this combination strategy first groups models into clusters based on a distribution of past forecasting performance, attributes forecasts to each cluster, and then estimates the optimal combination weight for each model. The mathematical procedure of this method is exorbitantly intricate and out of the purview of this study. Additionally, the outcome of this method shows strong robustness.

## 2.3 Criteria

Three measurements criteria [27], mean absolute error (MAE), mean absolute percentage error (MAPE) and root mean square error (RMSE), are used to evaluate the goodness-of-fit of the models and are defined as follow:

$$\text{MAE} = \frac{1}{n} \sum\nolimits_{i=1}^{n} |F_i - A_i| \tag{11}$$

$$\text{MAPE} = \frac{1}{n} \sum\nolimits_{i=1}^{n} \left| \frac{F_i - A_i}{A_i} \right| \times 100\% \tag{12}$$

$$\text{RMSE} = \sqrt{\frac{1}{n} \sum\nolimits_{i=1}^{n} (F_i - A_i)^2} \tag{13}$$

where n is the total number of data points, and $F_i$ and $A_i$ denote the predicted value and the actual value for the ith data point, respectively.

## 3. Results

Fig 1 visually demonstrates the 186 monthly cases of HFRS in mainland China ranging from January 2004 to June 2019. The preliminary plot illustrates the obvious seasonality pattern (s = 12) inherent in the incidence data for the HFRS disease. The incidence peaks twice every year with almost the same period. The prevalence of HFRS has a slight trend of diminishing in recent years. Fig 2 further vividly verifies the high significance of seasonality.

The ACF test and PACF test (Fig 3) were employed to determine the parameters of the ARIMA model. With the assistance of the residual test, we obtained six appropriate models as well as their AIC values. The AIC values of ARIMA$(0,1,3)(1,1,1)_{12}$, ARIMA$(0,1,3)(1,1,0)_{12}$, ARIMA$(0,1,3)(0,1,1)_{12}$, ARIMA$(3,1,0)(1,1,1)_{12}$, ARIMA$(3,1,0)(1,1,0)_{12}$ and ARIMA$(3,1,0)$ $(0,1,1)_{12}$ were -142.51, -143,11, -141.60, -141.56, -143.56 and -142.44, respectively. The ARIMA$(3,1,0)(1,1,0)_{12}$ model had the lowest AIC value and was chosen as the best model in the ARIMA approach. Its coefficients are shown in Table 1. The Ljung-box test showed the absence of autocorrelations (Fig 4) within the residuals.

Table 2 shows the fit and prediction results of the six models, and the accuracy rates are calculated by using the RMSE, MAE and MAPE. According to the results of the three criteria, the model stacking method outperformed all other approaches in terms of fitting accuracy and prediction accuracy. From the perspectives of the single model approaches, it is noteworthy that a model that functions well in terms of fit does not necessarily produce satisfactory prediction results. The ARIMA model has the worst performance in terms of both fitting and prediction. From the perspective of model stacking, Fig 5 shows the respective weights of the five models when factoring them into the model stacking procedure. The STL model has the highest weight, and the ARIMA model has the lowest weight. Fig 6 compares the mean forecasts of the six approaches. The NNAR model showed the largest deviation from the test set data, and the other five models performed relatively well with regard to prediction.

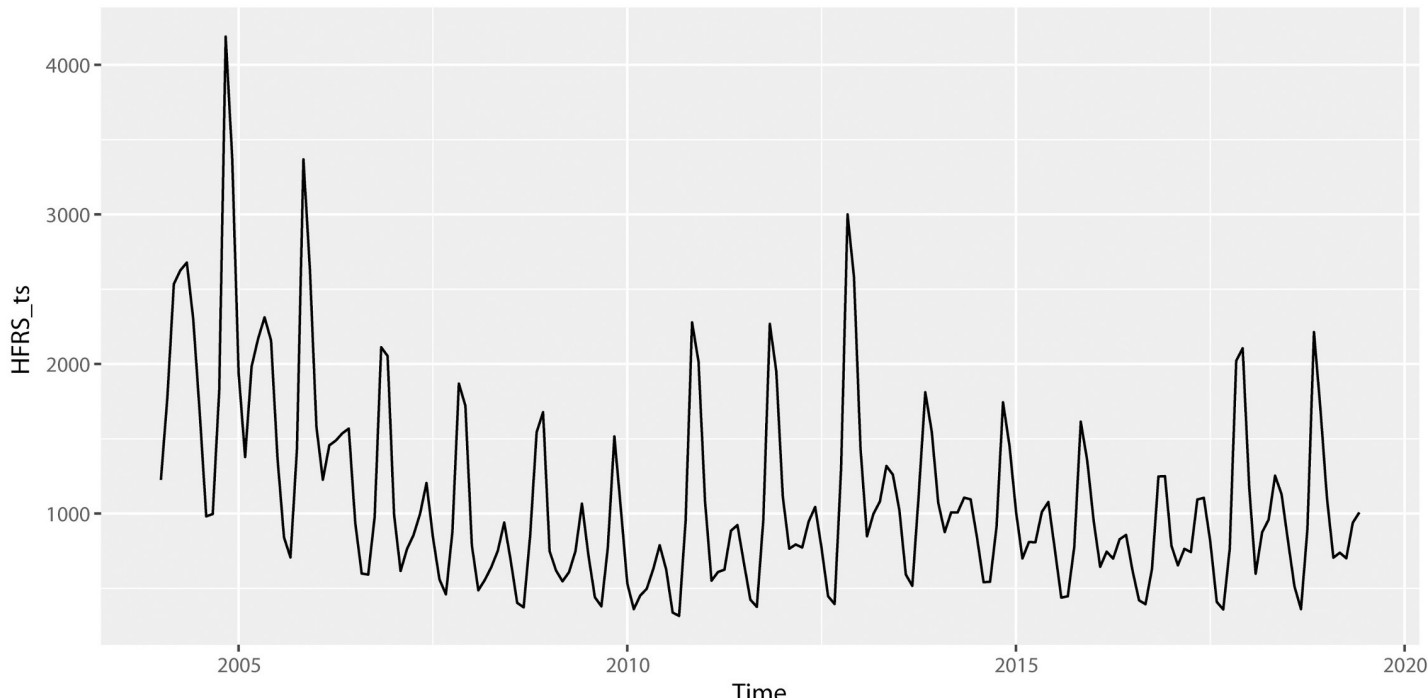

**Fig 1. Monthly cases of HFRS from January 2004 to June 2019 in mainland China.**

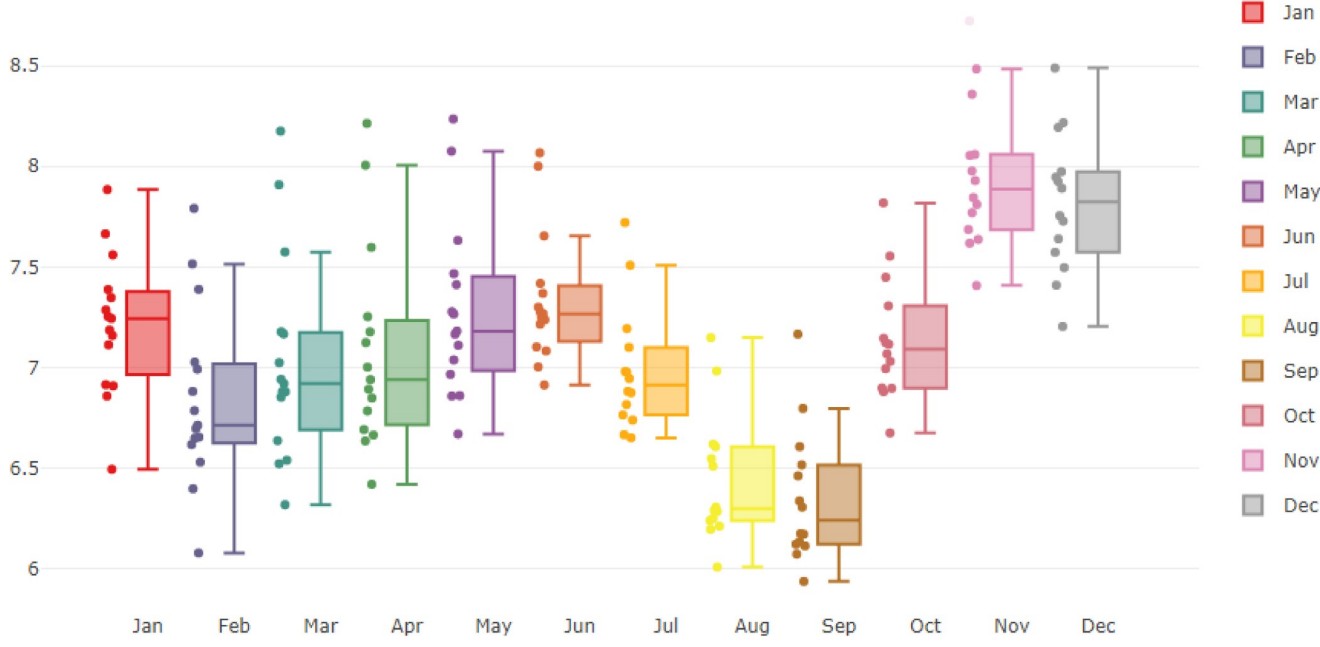

**Fig 2. The seasonal component of the monthly incidence data for HFRS.**

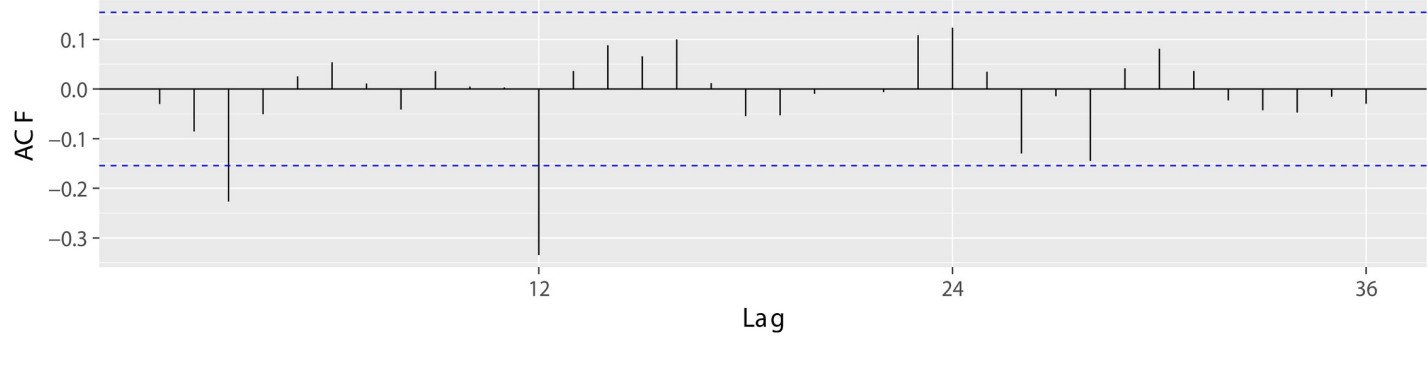

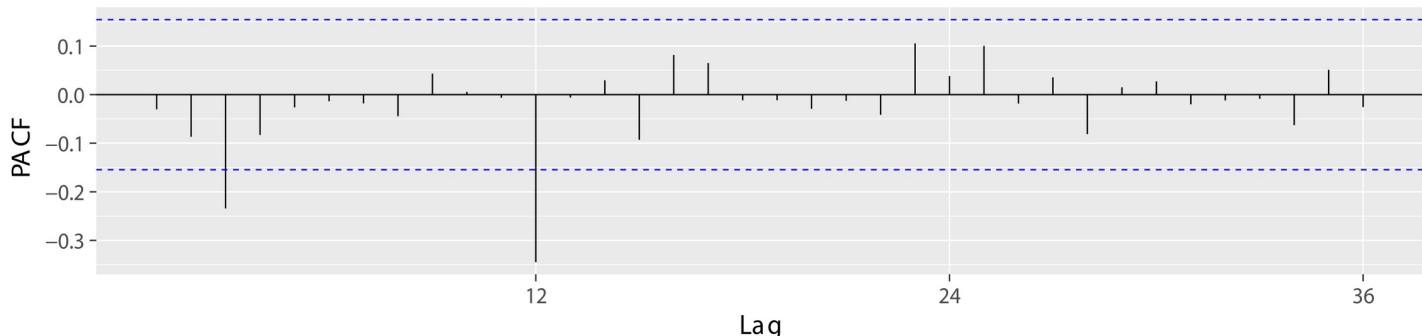

**Fig 3. ACF and PACF results obtained after differencing.**

**Table 1. Sparse coefficients of ARIMA(3,1,0)(1,1,0)$_{12}$.**

|  | Coefficient | 95%CI |
|---|---|---|
| AR1 | NA | NA |
| AR2 | NA | NA |
| AR3 | -0.29 | -0.45~-0.13 |
| SAR1 | -0.45 | -0.69~-0.29 |

## 4. Discussion

Statistical approaches have been believed to be an important part of infectious disease control and prevention in modern society for years. In this study, we constructed five models (ARIMA, STL, HW, NNAR, TBATS) and then combined their forecast data (means) based on their optimal weights to obtain the best predicted result; we also determined their accuracies of fit and forecasting via several criteria. The final results showed that model stacking is the best approach for fitting and predicting the monthly incidence of HFRS in mainland China, and this could potentially be a better tool for helping people make public health policies than existing methods.

The ARIMA model is a common method often used in infectious disease prediction, including the prediction of HFRS, and it has proven to be efficient in many studies. The ARIMA model is suitable for time series data with long-term trends and significant seasonal periodicity. As shown in Fig 1, the incidence of HFRS had a slightly downward trend, it usually reached its peak during winter, and there was another much smaller peak during autumn [28–30]. This phenomenon is probably related to rodent behavior, which is correlated with weather conditions, such as temperature. In this study, the ARIMA model was deployed as a conventional baseline model to evaluate the performances of the others approaches [31, 32].

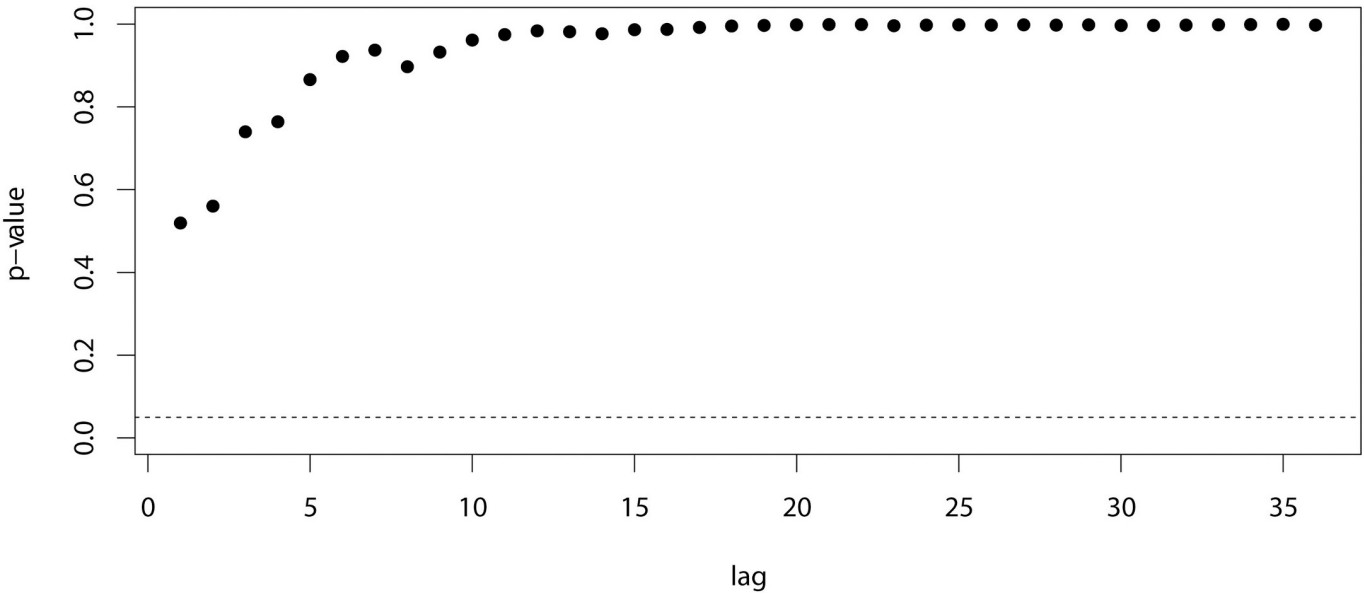

**Fig 4. Ljung-Box test of ARIMA(3,1,0)(1,1,0)$_{12}$.**

**Table 2. Accuracies of fit and prediction for six approaches.**

| Model | Training set | | | Test set | | |
|---|---|---|---|---|---|---|
| | RMSE | MAE | MAPE | RMSE | MAE | MAPE |
| ARIMA | 182.741 | 110.459 | 10.459 | 215.522 | 167.848 | 18.074 |
| HW | 151.313 | 104.103 | 9.788 | 152.511 | 125.459 | 14.907 |
| STL | 136.487 | 90.4280 | 8.461 | 154.758 | 131.556 | 15.624 |
| NNAR | 149.249 | 112.972 | 11.775 | 227.803 | 140.396 | 13.533 |
| TBATS | 150.894 | 104.325 | 9.829 | 154.082 | 129.137 | 14.883 |
| Combination | 128.190 | 85.631 | 8.118 | 151.864 | 118.273 | 13.156 |

The STL model can be applied to decompose time series without being limited to any certain kind of seasonality or periodicity. It allows its seasonal component to change over time and is more immune to outliers than other decomposition approaches. The STL method is gifted with the meritorious characteristics of versatility, robustness, and high applicability. The HW seasonal method is utilized to capture the seasonality of time series data by assuming that the predicted data and historical data share some common iterative data-generation patterns, namely, the estimation of future values is based on past data. It is comprised of three components: level, trend and seasonality. By dividing a time series into three sections and fitting them with their respective equations, the HW seasonal method can achieve good performances for time series with trends and seasonality. The NNAR model is a simple three-layer feedforward neural network with only one hidden layer. A neural network is gifted with a unique and excellent ability in that it can mine and capture the dynamics within a dataset by applying weights automatically (such as in the NNAR model), without sophisticated prerequisites and conditions. When applying weights to each function in its algorithm, this procedure is automatic and random. Therefore, we randomly applied weights 1000 times and then counted the mean of those outcomes. Based on this protocol, we ran the adjusted NNAR model 100 times, its outcomes maintained significant stability. The TBATS method is a conglomerate of the trigonometric seasonal model, a Box-Cox transformation, ARIMA and decomposition. These components are organically and coherently combined to fit time series data. The mechanism of TBATS is mathematically complicated, but it has been proven to be robust and applicable to time series data.

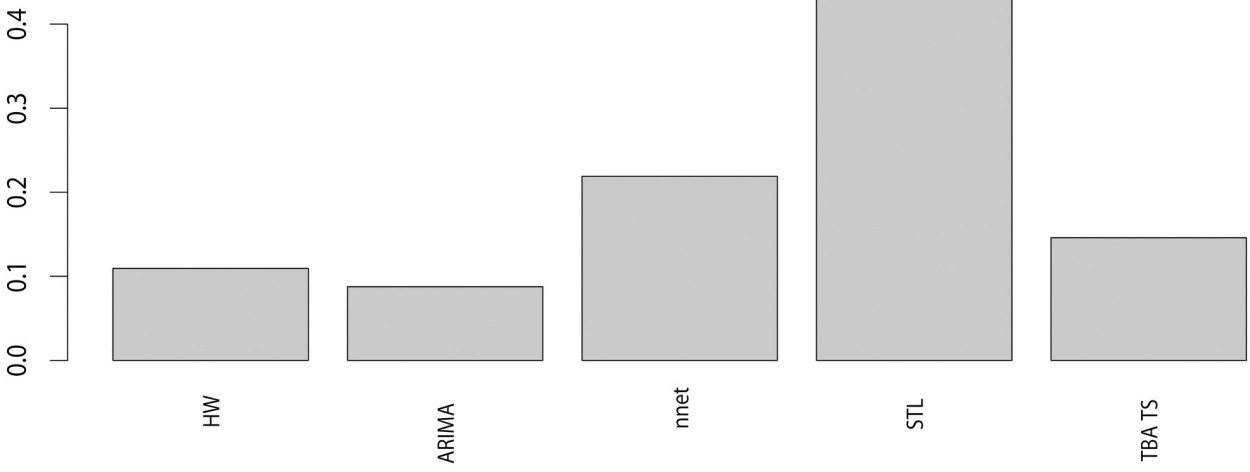

**Fig 5. The weight of each model as a combination of means.**

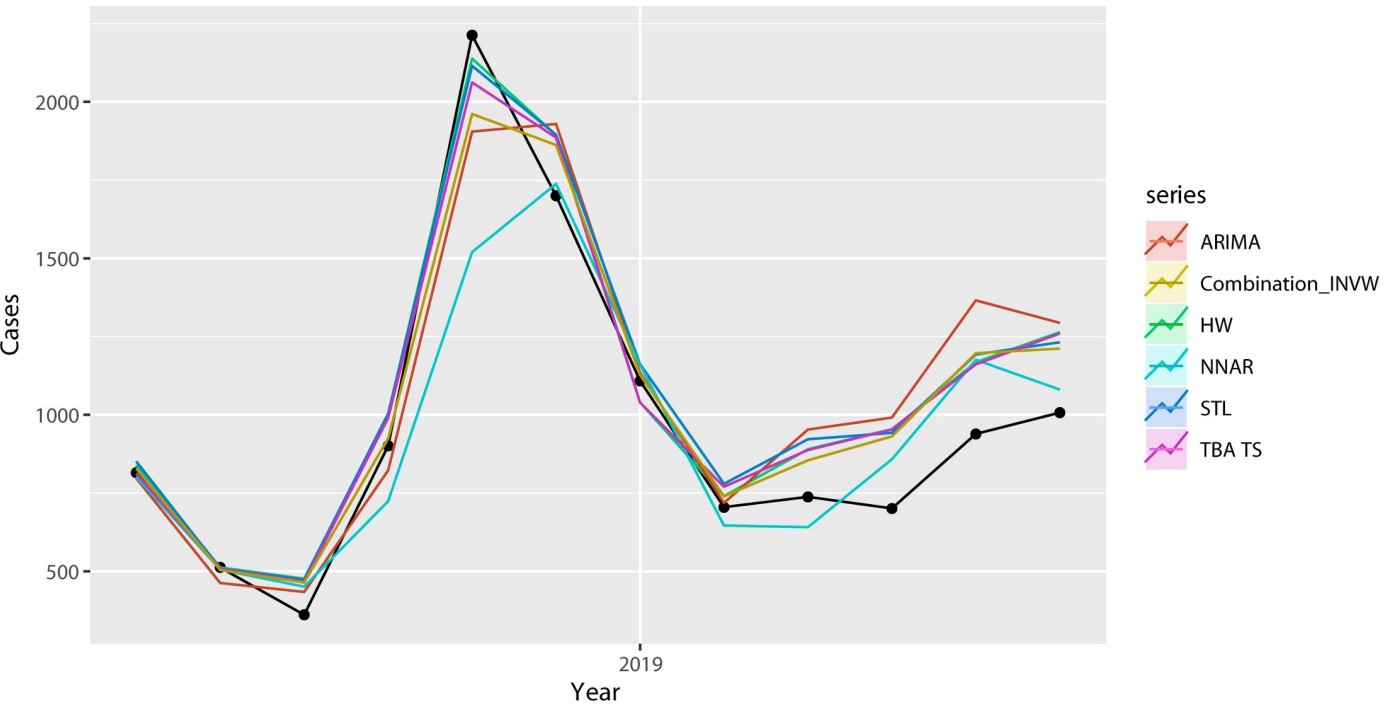

**Fig 6. Visual comparison of the predictions of five approaches.**

In contrast with model stacking, there is a hybrid model approach that is often used to enhance the accuracy of disease prediction, where researchers combine several models together to address the instability of a one-model approach. Such as, Wang YW et.al. [33] constructed a hybrid ARIMA-GRNN model to fit the monthly incidence of HFRS in mainland China, and this hybrid model outperformed the single ARIMA and GRNN models. And Wu W et al. [10] found that the hybrid ARIMA-NARNN model is an effective method to better fit the dynamics and characteristics of HFRS than the single model approach. Ling sun et al. [34] successfully applied a geographical information system together with the SARIMA model to denote and forecast HFRS epidemics with the goal of surveillance and control of HFRS in mainland China in a national level. Chang Qi et al. [35] compared the differences in fitting and prediction accuracies of SARFIMA and SARIMA when applied to HFRS disease and found that the SARFIMA model outperforms the SARIMA model in terms of improving the forecast of monthly HFRS cases. Youlin Zhao et al. [36] demonstrated that the SD-STARIMA model offers noticeably better prediction accuracy than the traditional approach, especially for spatiotemporal series data with seasonality characteristics.

Compared with those studies, we combined the prediction results of different models instead of using a hybrid model beforehand. Despite some essential differences between these approaches, they have been both proven to improve forecasting accuracy to a certain extent. These five base models can be theoretically categorized as linear models, nonlinear models and decompositions of sequences. The components that we input into the model stack are diverse, and the diversity of a single model is helpful in enhancing the accuracy of fitting and prediction.

## 5. Conclusion

In this paper, we fit monthly incidence data with five different models, and then we combined the forecasting results of the models by their weighted means to compare the performances of

a single model and model stacking. In this scenario with HFRS incidence data, we can safely conclude that combining multiple models leads to a significant improvement in accuracy. The model stacking approach outperforms any other one-model approach. The stacking of different models for the same time series data can be potentially utilized as a simple way to enhance the precision of forecasting infectious diseases, and this would more or less facilitate the prevention and control of certain diseases by preemptively acknowledging their characteristics.

## 6. Strengths and limitations of this study

Ensemble machine learning approaches are rarely used in HFRS disease prediction. Through this study, we can safely conclude that model stacking produces better outcomes than using a single model. There were several limitations of this study that should be pointed out. This study was an ecological study. There are many other factors, such as weather conditions and human activities that are related to the incidence of HFRS, but in our study, we built the model with only incidence data, thereby excluding many other potentially important data. A reliable and accurate forecasting model can help optimize resource allocation and mitigate the effects of epidemic outbreaks. In addition, the models we used in this study are limited and may not be the optimal model combination in the case of HFRS. The number and category of models we used to fit the data need further revision to optimize the results. Due to the lack of diversity in the HFRS data, we only developed a model for short-term forecasting to lower the deviation and error of the results. To improve the performance of our method, we need more data of other kinds to fit into the model, and we should revise the model as well. At the same time, our data we collected at the national level, so the applicability and effect of the model stacking method with regard to HRFS at the provincial or municipal level demand further study. Furthermore, HFRS is widely endemic in China and is mainly caused by two main genotypes of Hantaviruses, Hantaan and Seoul viruses, which have different seasonality and usually have different geographical distributions. The infection peaks of Hantaan virus mostly occurred in autumn-winter and spring, and the Seoul virus epidemic mainly occurred in spring. The epidemic dynamics usually varied in different endemic areas, and this study fitted and forecasted the monthly occurrence rate of HFRS at the national level, so it is likely possible that this study neglected the epidemic heterogeneity in different endemic areas.

## Supporting information

**S1 Dataset. Monthly cases of hemorrhagic fever with renal syndrome in mainland China from January 2004 to June 2019.**
(CSV)

## Author Contributions

**Conceptualization:** Peng Guan, De-sheng Huang, Bao-sen Zhou, Wei Wu.

**Data curation:** Guo-hua Ye.

**Formal analysis:** Bao-sen Zhou, Wei Wu.

**Funding acquisition:** Peng Guan, De-sheng Huang, Bao-sen Zhou, Wei Wu.

**Investigation:** Mirxat Alim, Bao-sen Zhou, Wei Wu.

**Methodology:** Peng Guan, De-sheng Huang, Bao-sen Zhou, Wei Wu.

**Resources:** Mirxat Alim, Wei Wu.

**Software:** Guo-hua Ye, Mirxat Alim.

**Supervision:** Mirxat Alim.

**Validation:** Guo-hua Ye, Mirxat Alim.

**Visualization:** Guo-hua Ye.

**Writing – original draft:** Guo-hua Ye.

**Writing – review & editing:** Guo-hua Ye.

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
