## [Decision Letter · Decision Letter 0]

4 Jan 2021

PONE-D-20-34402

Improving the precision of modeling the incidence of hemorrhagic fever with renal syndrome in mainland China with an ensemble machine learning approach

PLOS ONE

Dear Dr. Wu,

Thank you for submitting your manuscript to PLOS ONE. After careful consideration, we feel that it has merit but does not fully meet PLOS ONE’s publication criteria as it currently stands. Therefore, we invite you to submit a revised version of the manuscript that addresses the points raised during the review process.

We look forward to receiving your revised manuscript.

Kind regards,

Chi-Hua Chen, Ph.D.

Academic Editor

PLOS ONE

Journal Requirements:

2.PLOS requires an ORCID iD for the corresponding author in Editorial Manager on papers submitted after December 6th, 2016. Please ensure that you have an ORCID iD and that it is validated in Editorial Manager. To do this, go to ‘Update my Information’ (in the upper left-hand corner of the main menu), and click on the Fetch/Validate link next to the ORCID field. This will take you to the ORCID site and allow you to create a new iD or authenticate a pre-existing iD in Editorial Manager. Please see the following video for instructions on linking an ORCID iD to your Editorial Manager account: https://www.youtube.com/watch?v=_xcclfuvtxQ

<h1>** **</h1>

Reviewers' comments:

Reviewer's Responses to Questions

**Comments to the Author**

1. Is the manuscript technically sound, and do the data support the conclusions?

Reviewer #1: No

Reviewer #2: Yes

2. Has the statistical analysis been performed appropriately and rigorously? 

Reviewer #1: No

Reviewer #2: Yes

3. Have the authors made all data underlying the findings in their manuscript fully available?

Reviewer #1: No

Reviewer #2: Yes

4. Is the manuscript presented in an intelligible fashion and written in standard English?

Reviewer #1: No

Reviewer #2: Yes

5. Review Comments to the Author

Reviewer #1: 1. The abstract needs to be concise.

2. Try to improve the writing style.

3. Need to discuss about the novelty of the method proposed in this work.

4.Summarize the related work by adding table that should include aims of proposed work.

5.The proposed scheme is not clear at all along with a limited description.

6. Need to discuss about the used data and features in tabular form.

7.The novelty of the current paper is limited.

8. All mathematical equations need to numbered explicit in the number and its not just enough to write them.

Reviewer #2: This study aimed to fit an optimal to predict the HFRS incidence in China and proved that model stacking is an easy way to enhance prediction accuracy when modeling HFRS. The manuscript was well written, but some questions require revision and clarification.

Methods

1. Many machine learning methods, such as random forest (RF), support vector machine (SVM), long short term memory (LSTM) and so on, were employed to predict infectious diseases. Why did the authors select the five methods used in the manuscript rather than other methods?

2. Data from July 2018 to June 2019 were considered as test dataset. The data in the test set is insufficient.

3. In 2.3, the authors introduced 4 criteria to evaluate the performance of different models. However, MSE was not used in the results section.

4. The epidemic of HFRS is affected by many factors, such as meteorological factors, rodent behavior and so on. Why didn’t the authors fit multivariate models with related factors?

Results

1. In the first paragraph of the “Results”, what does the word “dtatfor” mean? Should it be revised as “data for”?

2. The authors said “the incidence peaks every other year” in the first paragraph of the “Results”. As shown in figure 1, however, the incidence presented two peaks in each year.

3. The author should briefly explain the main points expressed in Figure 2.

4. In 2.2.6, it should be explained how the weights of each model were calculated.

5. The fitting process of ARIMA model was expounded in details. But the modeling process of the other 4 models was not described.

6. The true values should be shown in Figure 6 to highlight the predictive performance of the different models.

Discussion

1. The content of the third paragraph in the “Discussion” is duplicated with that in the methodology section.

2. What’s the difference between the hybrid model and the stacking model used in this manuscript? It should be better to compare the performances of the hybrid model and stacking model predicting the HFRS incidence.

6. PLOS authors have the option to publish the peer review history of their article (what does this mean?). If published, this will include your full peer review and any attached files.

Reviewer #1: **Yes: **Dr Saurabh Pal

Reviewer #2: No

---

## [Author Response · Author response to Decision Letter 0]

16 Feb 2021

Reply to review 1:

1.The abstract needs to be concise.

Answer: we have reviewed the abstract.

2.Try to improve the writing style.

Answer: we have reviewed our writing style

3.Need to discuss about the novelty of the method proposed in this work.

Answer: in the introduction section of the manuscript, we have mentioned that ‘few studies have examined the combination of the outcomes of different models’ in term of predicting HFRS diseases, model stacking have been used in many studies of other realm, and the outcomes of the application of this method have been proved positive and useful, so we tried it on the prediction of HFRS disease in this study.

4.Summarize the related work by adding table that should include aims of proposed work.

Answer: the aims of this study pertaining to comparing the performance of six models have been included in the table 2. Our aims is to compare the accuracy of predicting and fitting among six models in a quantitative manner and we have tabulated it as table 2.

5.The proposed scheme is not clear at all along with a limited description.

Answer: we have mentioned the strategy in the abstract section, in our studies, We fitted the monthly reported cases of HFRS ranging from January 2004 to June 2019 in mainland China with an autoregressive integrated moving average (ARIMA) model; the Holt-Winter (HW) method, seasonal decomposition of the time series by LOESS (STL); a neural network autoregressive (NNAR) model; and an exponential smoothing state space model with a Box-Cox transformation; ARMA errors; and trend and seasonal components (TBATS), and we combined the forecasting results with the inverse rank approach (model stacking). Then we compared the performance of them in terms of fitting and predicting accuracy.

6.Need to discuss about the used data and features in tabular form.

Answer: the data we used doesn't include any feature, this study is a time series study without feature. The characteristics of the used data is illustrated in fig 1 and fig 2. The form of the used data is Monthly cases of HFRS from January 2004 to June 2019 in mainland China.

7.The novelty of the current paper is limited.

Answer: in the introduction section of the manuscript, we have mentioned that ‘few studies have examined the combination of the outcomes of different models’ in term of predicting HFRS diseases, model stacking have been used in many studies of other realm, and the outcomes of the application of this method have been proved positive and useful, so we tried it on the prediction of HFRS disease in this study.

8.All mathematical equations need to numbered explicit in the number and its not just enough to write them.

Answer: we have reviewed and numbered the equation. 

Reply to review 2:

Methods

1.Many machine learning methods, such as random forest (RF), support vector machine (SVM), long short term memory (LSTM) and so on, were employed to predict infectious diseases. Why did the authors select the five methods used in the manuscript rather than other methods?

Answer: because those five models are the most commonly used method in terms of HFRS time series study. That is why we have selected the five methods as our optimal selections, but that whether it is the best combination for model stacking or not hasn't been proved yet.

2.Data from July 2018 to June 2019 were considered as test dataset. The data in the test set is insufficient.

Answer: because the total of our data is limited, we had to limited the period of test dataset to one year to improve the accuracy of both fitting and predicting.

3.In 2.3, the authors introduced 4 criteria to evaluate the performance of different models. However, MSE was not used in the results section.

Answer: three criteria are enough to help us decide which model is the best in terms of predicting and fitting, so we have deleted MSE as our criteria.

4.The epidemic of HFRS is affected by many factors, such as meteorological factors, rodent behavior and so on. Why didn’t the authors fit multivariate models with related factors?

Answer: because this study is a time series study without considering any feature. It is not the aim of this study to add the meteorological factors as features. And some models of this study doesn't have the mechanism to include features as independent variables.

Results

1.In the first paragraph of the “Results”, what does the word “dtatfor” mean? Should it be revised as “data for”?

Answer: we have corrected the typo.

2.The authors said “the incidence peaks every other year” in the first paragraph of the “Results”. As shown in figure 1, however, the incidence presented two peaks in each year.

Answer: we have rephrased it as ‘ the incidence peaks twice every year’.

3.The author should briefly explain the main points expressed in Figure 2.

Answer: we have mentioned in the first paragraph in the result section that ‘ fig 2 further vividly verifies the high significance of seasonality’, and as well as ‘the incidence peaks twice every year”. that is the main points of fig 2.

4.In 2.2.6, it should be explained how the weights of each model were calculated.

Answer: the weights of each model are calculated automatically through the ‘inverse rank approach’ method. Briefly, this combination strategy first groups models into clusters based on a distribution of past forecasting performance, attributes forecasts to each cluster, and then estimates the optimal combination weight for each model. The mathematical procedure of this method is exorbitantly intricate and out of the purview of this study. 

5.The fitting process of ARIMA model was expounded in details. But the modeling process of the other 4 models was not described.

Answer: we haven explain briefly the mechanism of other models, the mathematical procedure of every details is intricate and out of the purview of this study. In practice, we just need to type the code of each model in R software, and we can get the result automatically.

6.The true values should be shown in Figure 6 to highlight the predictive performance of the different models.

Answer: we already compared the performance of each model in a concise and quantitative manner in table 2. the main point of fig 6 is to help me make sense of the difference of each model in an intuitive fashion.

Discussion

1.The content of the third paragraph in the “Discussion” is duplicated with that in the methodology section.

Answer: in the discussion, we feel the need to further detail the mechanism of each model and the advantage of each model by circumspectly discussing them together along with the characteristic of our data.

2.What’s the difference between the hybrid model and the stacking model used in this manuscript? It should be better to compare the performances of the hybrid model and stacking model predicting the HFRS incidence.

Answer: it is good point, but that missed the point of this study. The mechanism of hybrid model and model stacking are completely different. Hybrid model is to organically combine together, usually, two models, and it require that two models is able to connect each other. For example, in a hybrid model of ARIMA and neural network, we first fitted the raw data with ARIMA, after which we tried to obtain additional potentially available information from the residual with the help of a neural network. In this study, we fit the data with five different models and then combine the forecasting result by using the inverse rank approach to obtain the best possible weights of each model. So hybrid model method is not applicable to this study.

---

## [Decision Letter · Decision Letter 1]

2 Mar 2021

Improving the precision of modeling the incidence of hemorrhagic fever with renal syndrome in mainland China with an ensemble machine learning approach

PONE-D-20-34402R1

Dear Dr. Wu,

We’re pleased to inform you that your manuscript has been judged scientifically suitable for publication and will be formally accepted for publication once it meets all outstanding technical requirements.

Kind regards,

Chi-Hua Chen, Ph.D.

Academic Editor

PLOS ONE

Additional Editor Comments (optional):

Reviewers' comments:

Reviewer's Responses to Questions

**Comments to the Author**

1. If the authors have adequately addressed your comments raised in a previous round of review and you feel that this manuscript is now acceptable for publication, you may indicate that here to bypass the “Comments to the Author” section, enter your conflict of interest statement in the “Confidential to Editor” section, and submit your "Accept" recommendation.

Reviewer #1: All comments have been addressed

2. Is the manuscript technically sound, and do the data support the conclusions?

Reviewer #1: Yes

3. Has the statistical analysis been performed appropriately and rigorously? 

Reviewer #1: Yes

4. Have the authors made all data underlying the findings in their manuscript fully available?

Reviewer #1: Yes

5. Is the manuscript presented in an intelligible fashion and written in standard English?

Reviewer #1: Yes

6. Review Comments to the Author

Reviewer #1: The authors have addressed all the points raised by reviewer. Paper can be accepted for publication in present form.

7. PLOS authors have the option to publish the peer review history of their article (what does this mean?). If published, this will include your full peer review and any attached files.

Reviewer #1: **Yes: **Saurabh Pal

---

## [Editor Report · Acceptance letter]

5 Mar 2021

PONE-D-20-34402R1 

Improving the precision of modeling the incidence of hemorrhagic fever with renal syndrome in mainland China with an ensemble machine learning approach 

Dear Dr. Wu:

I'm pleased to inform you that your manuscript has been deemed suitable for publication in PLOS ONE. Congratulations! Your manuscript is now with our production department. 

Kind regards, 

on behalf of

Professor Chi-Hua Chen 

Academic Editor

PLOS ONE